# Application-Aware Resource Allocation Based on Benefit–Cost Ratio in Computing Power Network with Heterogeneous Computing Resources

Yahui Wang, Yajie Li *, Jiaxing Guo, Yingbo Fan, Ling Chen, Boxin Zhang, Wei Wang, Yongli Zhao and Jie Zhang

State Key Laboratory of Information Photonics and Optical Communications, Beijing University of Posts and Telecommunications (BUPT), Beijing 100876, China; yahui_wang@bupt.edu.cn (Y.W.); guojiaxing@bupt.edu.cn (J.G.); ying_bo@bupt.edu.cn (Y.F.); chen_0@bupt.edu.cn (L.C.); zbx2017@bupt.edu.cn (B.Z.)
* Correspondence: yajieli@bupt.edu.cn

**Abstract:** The computing power network (CPN) is expected to realize the efficient provisioning of heterogeneous computing power through the collaboration between cloud computing and edge computing. Heterogeneous computing resources consist of CPU, GPU, and other types of computing power. Different types of applications may have diverse requirements for heterogeneous computing resources, such as general applications, CPU-intensive applications, and GPU-intensive applications. Service providers are concerned about how to dynamically provide heterogeneous computing resources for different applications in a cost-effective manner, and how to deploy more applications as much as possible with limited resources. In this paper, the concept of the benefit–cost ratio (BCR) is proposed to quantify the usage efficiency of CPU and GPU in CPNs. An application-aware resource allocation (AARA) algorithm is designed for processing different types of applications. With massive simulations, we compare the performance of the AARA algorithm with a benchmark. In terms of blocking probability, resource utilization, and BCR, AARA achieves better performance than the benchmark. The simulation results indicate that more computing tasks can be accommodated by reducing 3.7% blocking probability through BCR-based resource allocation.

**Keywords:** computing power network; application-aware; heterogeneous computing resources; resource allocation





## 1. Introduction

Data processing requires powerful computing resources and extensive network connections that can function collaboratively across cloud, edge, and end. The computing power network (CPN) is a new network technology that can flexibly allocate and schedule computing resources, storage resources, and network resources among cloud, edge, and end [1,2]. It considers the network conditions and user requirements to provide the optimal distribution, association, transaction, and scheduling of resources [1]. The CPN can also publish the current computing power and network status to the network as routing information, and then the network routes the computing task to the corresponding computing nodes to achieve the best user experience and network efficiency [2]. Moreover, due to the dynamic nature of computing tasks, the tasks have the feature of high burstiness and high throughput in CPNs. Thanks to the tremendous bandwidth in fibers, optical networks have become the key infrastructure in CPNs. Optical transport networks (OTNs) with an agile bandwidth allocation have been developed to achieve large-capacity, low-latency, and high-efficiency performance [3]. Therefore, OTN has become a promising transport solution in CPNs.

CPNs offer heterogeneous computing resources to meet the requirements of various types of computing tasks, including CPUs, GPUs, and others [4–6]. The CPU is a common

processor that can handle a variety of computing tasks that require low latency and high reliability [7,8]. However, the GPU is a graphics processor that excels at high-speed floating-point calculations and parallel computing. Therefore, GPUs are particularly suitable for tasks that require high levels of parallelism [9]. To quantify the acceleration of GPU to data parallelism, the author in [10] presented a paleo-analysis performance model that simulated training the whole neural network through data parallelism. Typically, computing tasks can be calculated by software specification using a CPU or GPU. According to the application scenarios and requirements, we can divide the computing tasks into three types: (1) general applications (i.e., both CPUs and GPUs can be used) [11]; (2) CPU-intensive applications, which require significant CPU resources and have relatively low GPU demands, such as text parsing and semantic analysis in natural language processing [12]; and (3) GPU-intensive applications, which require significant GPU resources and have relatively low CPU demands, such as game design and image processing [13,14].

As a service provider, the allocation of computing resources is influenced by multiple factors, such as the Quality of Service (QoS) requirement and computing resource capacity. We need to consider the bandwidth, latency, and prioritization of different types of tasks simultaneously to ensure optimal user experience and satisfaction. The benefits of completing the computing task are also taken into account, since the service provider will be motivated by benefit goals. However, the benefits brought by fulfilling the task are not always proportional to the costs of computing resources [15,16]. In addition, resource utilization also needs to be considered, as service providers optimize the computing resource allocation to ensure the maximum efficiency of the resource usage [17]. Therefore, service providers are concerned about how to dynamically provide computing resources for multiple computing tasks in a cost-effective manner. The target is to minimize resource costs while maximizing the benefits of performing computing tasks within the constraints of the limited resources.

In order to adapt to the emerging computing requirements and different scenarios, researchers have made several attempts at the resource allocation of CPNs. Resource prediction can more accurately assess the execution time and resource requirements of tasks so as to allocate resources more effectively and avoid resource waste [18,19]. The authors in [20] proposed a dynamic idle interval prediction scheme. The scheme can forecast the CPU's idle interval length and choose the most cost-effective sleep state to achieve minimal power consumption. In addition, the reduction in computing resource costs and the enhancement of resource efficiency have progressively emerged as significant areas of study in CPNs. The author in [21] considered the server configuration with the lowest cost to meet time-varying resource demands in cloud centers. Resource allocation and the cost problem are formulated in [22] as a Stackelberg game, which not only maximizes the benefits of the provider, but also minimizes the costs of the services. Moreover, the author in [23] proposed a price bidding mechanism for multi-attribute cloud-computing resource provision from the perspective of a non-cooperative game. This mechanism aims to maximize the benefit of each player. However, the researchers failed to consider the provision of computing resources using a cost-effective method, and only focused on one type of computing resource (i.e., the CPU). To this end, we consider both CPU and GPU computing resources, and optimize the use of heterogeneous computing resources to reduce costs and increase benefits.

In this paper, we propose the concept of the benefit–cost ratio (BCR) to quantify the usage efficiency of CPUs and GPUs in CPNs. The benefits are obtained by accelerating the computing tasks through appropriate resource allocation. In general, the earlier completion of computing tasks ahead of deadline brings more benefits for service providers. Meanwhile, the costs are generated by occupying bandwidth and computing resources for tasks. An application-aware resource allocation (AARA) algorithm is designed for processing applications based on BCR in CPNs with CPUs and GPUs. The main contribution of the AARA algorithm is to provide cost-effective computing resources for computing tasks, while accommodating more applications by efficiently utilizing heterogeneous re-

sources. We compare our algorithm with an application-unaware resource allocation (AURA) algorithm. Our performance metrics include CPU utilization, GPU utilization, bandwidth utilization, blocking probability, and BCR. The results show that the proposed AARA algorithm can significantly improve the BCR while improving the efficiency of resource utilization.

The remainder of the paper is organized as follows. In Section 2, we describe the problems of resource allocation in CPNs. The network model and AARA algorithm are proposed in Section 3. Section 4 describes the performance evaluation of the proposed algorithm using simulation results. In Section 5, we discuss the limitations and improvement directions. The conclusions are described in the last section.

## 2. Problem Statement

### 2.1. Applications in CPNs with Heterogeneous Resources

The CPN integrates computing resources into the communication network, offering users the most suitable computing resource services in a more holistic form. Figure 1 shows the composition of the CPN, including the computing layer where CPUs and GPUs supply computing resources, the IP layer where the Ethernet Switch (E-Switch) aggregates traffic, and the optical layer where a ROADM provides wavelength bypass and switching [24]. The E-Switch is a common device used to connect multiple network devices and facilitate the forwarding and switching of data packets. Additionally, the computing node is equipped with multiple high-performance servers, where CPUs and GPUs are integrated to provide computing power for applications. These high-performance servers are connected to an E-Switch to facilitate data exchange and interaction among multiple servers. Meanwhile, the E-Switch is connected to a ROADM to convert electrical signals into optical signals for data transmission.

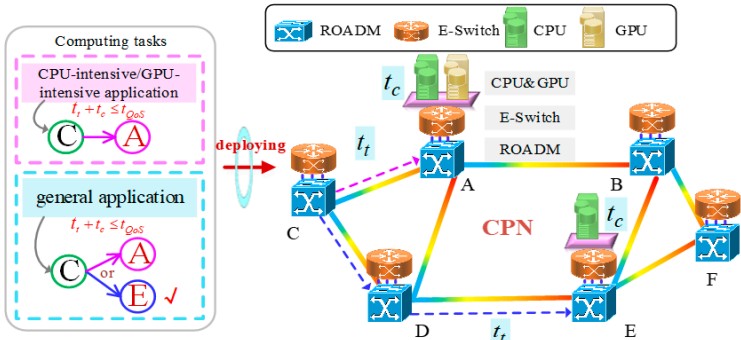

**Figure 1.** Applications in CPN with heterogeneous resources.

In Figure 1, we propose a heterogeneous CPN model composed of nodes and links with a certain capacity, where the nodes are divided into computing nodes and transmission nodes. Computing nodes refer to the nodes connected to computing resources, while transmission nodes are only for data transmission. Note that all computing nodes have CPU resources, while only a portion of the nodes have GPU resources. In Figure 1, node A is a computing node with CPU and GPU, node E is a computing node with CPU, and nodes B, C, D, and F are all transmission nodes. In addition, we illustrate the three mentioned types of applications. Specifically, general applications can be deployed on a computing node with either CPU or GPU resources, but it is crucial to choose a node with sufficient resources to accommodate the application, such as node E. CPU-intensive and GPU-intensive applications require the selection of node A with CPU and GPU resources to satisfy the QoS requirements of the application. For successfully deployed applications, the sum of the transmission time $t_t$ and computing time $t_c$ must meet the QoS requirement $t_{QoS}$. Meanwhile, the applications are placed in the form of containers, considering the high efficiency of resource isolation and usage. In addition, the CPU and GPU resources re-

quired by the application are determined by the computational workload and computation completion time.

## 2.2. Resource Allocation for Applications

The service provider provides computing resources for an application and therefore needs to consider the benefit of an application [16]. However, the benefit of an application is influenced by various factors, such as network resources, computing resources, and resource costs. Specifically, the relationship between benefit and cost is not always proportional, and the related benefits of applications may increase non-proportionally with the increase in the resource cost. Additionally, user experience and satisfaction should also be considered [25]. If an application can provide a good user experience, it will result in greater user loyalty and bring more benefits to the service provider. Thus, service providers need to comprehensively consider these factors to measure the benefit brought by an application and select a resource allocation scheme.

Figure 2 illustrates the problem of resource allocation for applications in CPNs. Figure 2a shows the problem of service providers allocating resources for different applications. We use grades to describe the values of computing power parameters, where different grades represent different levels of computing resource consumption. Therefore, when a service provider allocates computing power grade $j$ to application $i$, it incurs corresponding $costs(i, j)$ and generates the $benefits(i, j)$ in this computing power grade. Figure 2b shows the trend of the benefit–cost ratio for allocating computing resources to the application. The trend indicates that the benefits of the application and the costs of resource investment exhibit a nonlinear relationship.

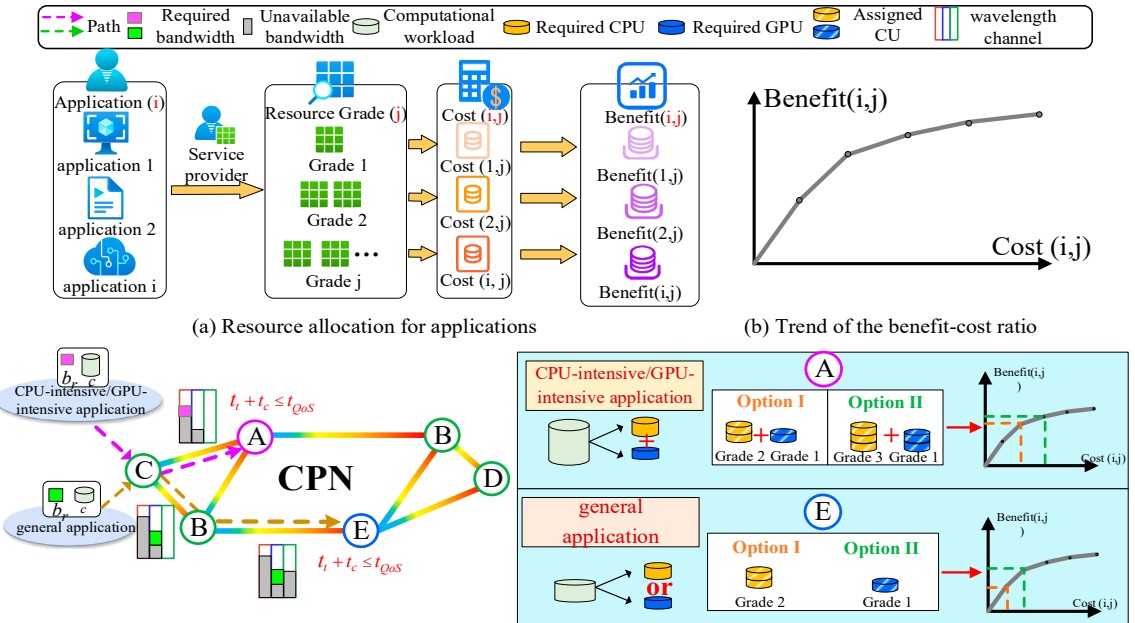

**Figure 2.** Illustration of resource allocation for application.

Figure 2c demonstrates different options of resource allocation for three types of applications, where $b_r$ and $c$ represent the bandwidth requirements and computational workload of the applications, respectively. We assume that the CPU-intensive/GPU-intensive application is initiated from source node C and deployed in computing node A, while the general application is deployed in computing node E. Meanwhile, the deployed path of both applications satisfies the bandwidth requirements. To meet the QoS requirements, we allocate sufficient computing resources to the applications. The right side of Figure 2c presents two different options of computing resource allocation for the mentioned applications. From the example, it can be observed that allocating more computing resources does not necessarily

result in proportional increases in application benefits. For CPU-intensive/GPU-intensive applications at node A, option II allocates more computing resources compared to option I, but the benefit only increases slightly. However, for the general application deployed in node E, the higher cost of resource investment in option II also brings higher benefits. Therefore, how to allocate resources for applications using a cost-effective method is a worthwhile problem to research.

## 3. Application-Aware Resource Allocation Model

In this section, we propose the network model and application-aware resource allocation (AARA) algorithm.

### 3.1. Network Model

In this paper, we approach the allocation of computing resources for three types of applications, as well as taking into account the impact of latency on application QoS and its associated benefits. The concept of the BCR is proposed to quantify the usage efficiency of the CPU and GPU in CPNs. Figure 3 shows the network model and AARA algorithm for different applications. The types and grades of computing resources in node A with CPU and GPU and node E with CPU are indicated in Figure 3a. We classify the CPU and GPU resources of computing nodes into different grades, where each grade represents a specific size of computing resource. Meanwhile, a higher grade of computing resources corresponds to a higher cost. The task is denoted as $i\{\mu, m, s, c, b_r, t_b, t_e, t_{QoS}\}$, where $\mu$ is the application priority, $m$ is the application type, $s$ is the source node, and $d$ is the data size. The computational workload and bandwidth required for the computing task are denoted as $c$ and $b_r$, respectively. Moreover, $c$ consists of $c_{CPU}$ and $c_{GPU}$, $t_b$ is the application begin time, $t_e$ is the end time, and $t_{QoS}$ is the delay requirement of the QoS.

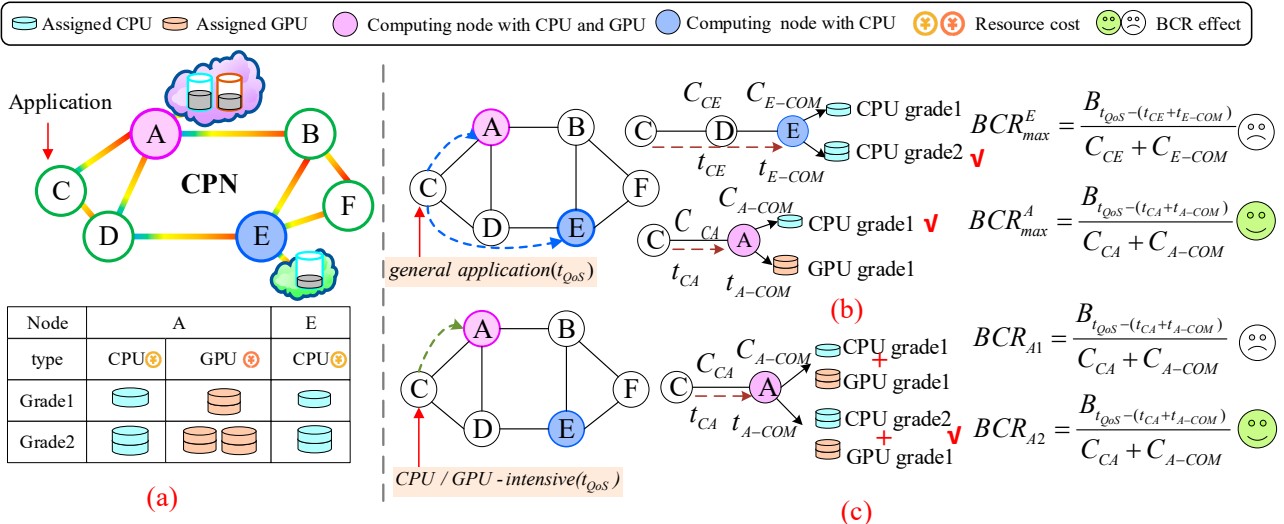

**Figure 3.** Network model (**a**) CPN with heterogeneous resources. (**b**) AARA for general application (**c**) AARA for CPU/GPU-intensive application.

Figure 3b shows an example of the BCR-based resource allocation for general applications. The general applications can select nodes A and E for computing when the bandwidth requirements are satisfied. We provide four grades of computing power available for general application in nodes A and E. Then, we determine the deployed cost and generated benefit for each grade of computing power, as well as find the optimal BCR of different grades in computing nodes A and E to deploy the general application. We define cost as the consumption of computing and network resources by deploying the application, calculated using Equation (2). The generated benefits of the application are related to latency and calculated using Equation (3). Finally, we select the CPU grade 1 of node A for general application. In Figure 3c, we provide an example of resource allocation

for CPU/GPU-intensive applications. Based on the type of resource requirement for the application, the computing node A with a CPU and GPU is chosen. The link between nodes C and A satisfies the bandwidth requirement. Then, the required CPU and GPU resources are allocated in proportion and two options of computing power grades can be selected. Finally, the CPU grade 2 and GPU grade 1 are assigned for CPU/GPU-intensive applications in comparison to $BCR_{A1}$ and $BCR_{A2}$.

In this paper, we define $BCR$, namely, the benefit–cost ratio. Our objective is Equation (1), where $R$ is the set of applications.

$$Max\sum_{r} BCR, r \in R\{1,2,3\ldots,r\} \tag{1}$$

$BCR$ is calculated in Equation (2), where $B$ is benefit, $C_{com}$ is computing resource cost, and $C_t$ is the bandwidth resource cost consumed by the computing tasks.

$$BCR = \frac{B}{C_{com} + C_t} \tag{2}$$

The benefit is formulated as in Equations (3)–(7).

$$B = \mu \times T \times ln(1 + (t_{QoS} - t_f)) \tag{3}$$

$$t_f = t_t + t_{com}, t_f \leq t_{QoS} \tag{4}$$

$$T = t_e - t_b \tag{5}$$

$$t_t = D \cdot v \tag{6}$$

$$t_{com} = \frac{c}{CU_\tau^l} \tag{7}$$

In Equation (3), $\mu$ is the priority of the computing tasks [26], $T$ is the duration of the application, and $t_{Qos}$ is the delay requirement of QoS [27]. The actual processing time $t_f$ is calculated using Equation (4). In addition, the duration $T$ follows an exponential distribution. In Equation (5), $t_e$ is the end time and $t_b$ is the begin time. Meanwhile, in Equation (6), D is the distance from source node to computing node and $v$ is the transmission speed. Finally, the computing time is calculated using Equation (7), where $CU_\tau^l$ is the computing unit capacity of different grade $l$, $l$ is the grade number, and $\tau$ is the type of computing resource, either CPU or GPU.

The cost is formulated as in Equations (8) and (9).

$$C_{com} = \alpha \cdot \frac{CU_{CPU}}{CU_{CPU\_min}} \cdot \theta + \beta \cdot \frac{CU_{GPU}}{CU_{GPU\_min}} \cdot \delta \tag{8}$$

$$C_t = b_r \cdot D \cdot \varepsilon \tag{9}$$

Equations (8) and (9) are used to calculate the cost, which is divided into two parts, namely, computing resource cost and network resource cost. Equation (8) represents the computing resource cost, where $\alpha$ and $\beta$ denote the ratio of required CPU and GPU, $\theta$ and $\delta$ represent the unit cost consumed by each grade of resources. Equation (9) is the network resource cost, where D is the distance from the source node to computing node, and $\varepsilon$ is the cost value per Gbps·km [26].

The average BCR of applications is defined as:

$$BCR_{ave} = \frac{\sum_{r=1}^{v} BCR_r}{N_S} \tag{10}$$

where $N_S$ is the number of applications that are successfully deployed in the computing nodes. This paper attempts to obtain the optimal application average BCR with resource constraint by deploying applications in a cost-effective manner.

### 3.2. AARA Algorithm

As shown in the pseudocode provided in the Algorithm 1, we first determine the type of application and subsequently select the appropriate computing node based on the required computing resources types. Then, we search for a set of paths $P$ from the source node to the destination computing node, and calculate the BCR for each computing node that meets the computing requirements. Finally, we select the deployment scheme with the maximum BCR for application.

Some applications generate small benefits, but occupy a large amount of computing resources, resulting in resource wastage, while certain applications generate significant benefits, but lack sufficient computing resources. Therefore, we used a baseline algorithm called the AURA to compare our algorithm. Additionally, the term unaware refers to the lack of awareness regarding the application's corresponding benefits. The main difference between the AURA and AARA algorithms is that AURA only provides a single grade of resources based on the requirements without considering the benefits or BCR of each application. Specifically, AURA selects suitable computing nodes for different application types to meet their requirements first. Then, it searches for a set of paths $P$ using the K-Shortest Path (KSP) algorithm. The application will be blocked when the $P$ is empty. Meanwhile, if there is a shortage of computing resources, the application will also be blocked.

---

**Algorithm 1: AARA**

---

**Input**: Give $G_P = \{V_P, E_P\}$, $e \in E_p$, $n_i \in V_P$;
$r = \{(\mu, m, s, b_r, c, t_b, t_e, t_{QoS})\}$, $\mu \in [1, 4]$, m $\in \{\text{general}, \text{CPU\_intensive}, \text{GPU\_intensive}\}$; grade
$G = \{CU_\tau^l\}$, $l \in [1, 4]$, $\tau \in \{\text{CPU}, \text{GPU}\}$

---

**Output**: $CU_\tau^l$, $<P, n_i>$

---

Initialized system computing nodes and network parameters;
1. ***While*** there are undeployed applications ***do***
2.   ***for*** each undeployed application $r$ ***do***
3.     ***if*** m is CPU/GPU-intensive ***then***
4.       Distribute $c$ proportionally for CPU and GPU, $\{c_{cpu}, c_{gpu}\}$
5.       ***for*** each computing node $n_i$ ***do***
6.         Exclude node with insufficient computing resource
7.         ***for*** each sufficient computing node $n_i$ ***do***
8.           Search for path set $P$ can satisfy $b_r$ by using KSP algorithm
9.           ***if*** $P$ is the empty set ***then*** blocking
10.           Search for grades set $G = \{CU_\tau^l\}$
11.           ***for*** $g$ in $G$ ***do***
12.             Use Equations (1)–(8) compute *BCR*
13.           ***end for***
14.         ***end if***
15.         ***end for***
16.       ***end for***
17.     ***end if***
18.     ***else if*** m is general ***then***
19.       Compute CPU or GPU computing resource required
20.       The same as Step 5–16
21.     ***end if***
22.   Find a computing node $n_i$ and grade $CU_\tau^l$ with the max BCR;
23.   Deploy application to computing node $n_i$
24.   Update computing node $n_i$ and network parameters.
    Remove tasks $r$ from undeployed tasks set $R$
25.   ***end for***
26. ***end while***
27. Use Equation (10) to compute $BCR_{ave}$.

---

## 4. Simulation Setup and Results

### 4.1. Simulation Setup

To evaluate the effect of our work, as shown in Figure 4, we use an NSFNET topology consisting of 14 optical nodes (of which 5 computing nodes and 9 transmission nodes) and 22 fiber links. We consider the computing tasks as the form of data flow, and the arrival time and duration time of the computing tasks follow Poisson distribution and exponential distribution, respectively. For the same source node and destination node, multiple computing tasks can be aggregated into a single wavelength channel for data transmission to improve network resource utilization.

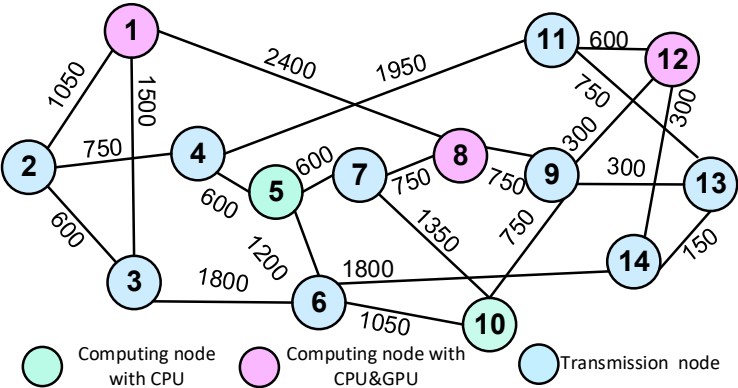

**Figure 4.** NSFNET topology.

The specific parameters are shown in Table 1, where $V_h$ is the set of computing nodes with CPU and GPU, $V_c$ is the computing nodes with CPU, $C_{CPU}$ and $C_{GPU}$ are CPU and GPU capacity of computing node, respectively [28], $N_W$ is the number of wavelength channels [29], b is the bandwidth capacity of each wavelength [3], and $v$ is the transmission speed [30]. In addition, the proportion of general, CPU-intensive, and GPU-intensive applications is $r_g$, $r_C$, and $r_G$, respectively.

**Table 1.** Parameter setup in simulation.

| Category | Parameters | Value |
|---|---|---|
| Network parameters | $V_h$ | [1, 8, 12] |
| | $V_c$ | [5, 10] |
| | $C_{CPU}$ | 4TFLOPS |
| | $C_{GPU}$ | 8TFLOPS |
| | $CU^l_{CPU}$ | [50, 100, 150, 200] GFLOPS |
| | $CU^l_{GPU}$ | [100, 200, 300, 400] GFLOPS |
| | $CU_{CPU}$ | 50GFLOPS |
| | $CU_{GPU}$ | 100GFLOPS |
| | $N_W$ | 80 |
| | b | 100 Gbps |
| | $v$ | 5 μs/km |
| Computing task | $\mu$ | [1, 2, 3, 4] |
| | c | $[4, 20] \times 10^8$ FLOPs |
| | $b_r$ | [3, 8] Gbps |
| | $t_{QoS}$ | [10, 30] ms |
| | $r_g$ | 20% |
| | $r_C$ | 35% |
| | $r_G$ | 45% |
| | $\alpha : \beta$ | 1:4/4:1/1:3/3:1 |
| Cost value | θ | 0.2/(CUs·time unit) |
| | δ | 0.4/(CUs·time unit) |
| | ε | 0.00375/( Gbps·km·time unit) |

### 4.2. Results and Analysis

In this section, we analyze the performance of the proposed AARA algorithm under a limited resource and compare it to the performance of AURA in the same number of computing tasks. The effectiveness of the proposed algorithm is evaluated based on the CPU utilization, GPU utilization, bandwidth utilization, blocking probability, and BCR of the tasks. For each set of simulations, we define two cases for the workload ratio between the CPU and GPU of CPU-intensive and GPU-intensive applications. In Case 1, the ratio of CPU and GPU requirements for CPU-intensive and GPU-intensive applications is 1:3 and 3:1, respectively. In Case 2, these ratios are 1:4 and 4:1, respectively. The resource utilization ratios in the simulation results are all calculated under the traffic load of 450 Erlangs.

Computing resource utilization is a key metric in evaluating system efficiency and performance in CPNs. Figure 5 shows the results of the CPU utilization for 2000 and 3000 computing tasks. It can be observed that there are significant variations in resource utilization as tasks arrive and depart. When the number of tasks is 2000, our proposed AARA algorithm and the AURA algorithm achieve maximum resource utilization ratios with task ID between 1000 and 1500. The respective maximum CPU utilization ratios for the AARA and AURA algorithms are 66% and 62% in Case 1. In Case 2, the increased demand for CPU resources leads to higher CPU utilization. For task ID between 1000 and 1500, the maximum CPU utilization ratios for AARA and AURA are 86% and 78%, respectively. Due to the consideration of dynamic scenarios, resources are released when tasks depart, resulting in some fluctuations in resource utilization ratios. When the number of computing tasks is 3000, both AARA and AURA exhibit a peak CPU utilization ratio of 99% in Case 1 and Case 2. Furthermore, for both task numbers, AARA shows a higher peak of CPU utilization ratio compared to AURA. This can be attributed to the lower cost of CPU resources, which enables AARA to allocate resources with a higher benefit–cost ratio.

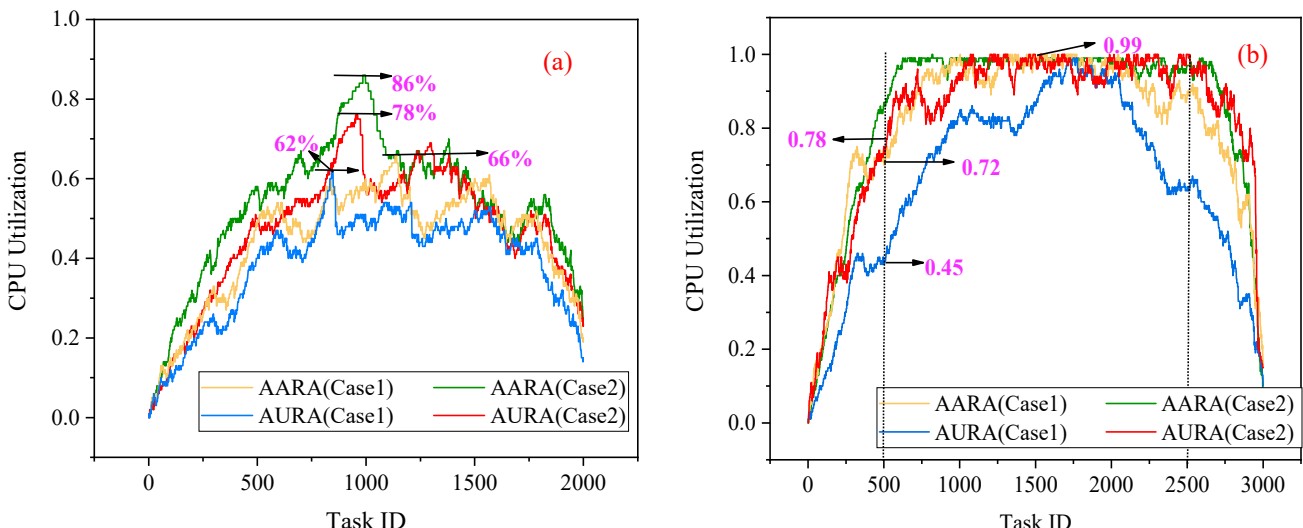

**Figure 5.** CPU utilization in different profiles of computing tasks: (**a**) 2000 tasks; (**b**) 3000 tasks.

Figure 6 shows the result of GPU utilization, in which the process of resource occupation and release coincides with the arrival and departure of tasks. Given the significant demand for GPU resources brought by AI, the simulation sets a larger proportion of GPU-intensive applications. Therefore, there is a higher demand for GPU resources. In Figure 6 a, when the workload ratio between the CPU and GPU is set as Case 1, for 2000 tasks, the GPU resource utilization reaches its peak of task ID between 750 and 1500. For AARA and AURA, the maximum GPU resource utilization ratios are 73% and 85%, respectively. When the workload ratio between the CPU and GPU is set as Case 2, the maximum GPU utilization ratios for AARA and AURA are 93% and 97%, respectively. When the number of tasks is 3000, the peak of the GPU utilization ratio for the AARA and AURA algorithms

are 99% in Case 1 and Case 2. From the CPU and GPU utilization ratios, it can be observed that the AARA algorithm shows higher CPU utilization, whereas the AURA algorithm shows higher GPU utilization. This is because CPUs are relatively lower in cost compared to GPUs, leading to a preference for CPU resources in the allocation of resources for general applications. Moreover, the AURA algorithm lacks awareness of the resource cost and benefit, leading to the inappropriate utilization of GPU resources.

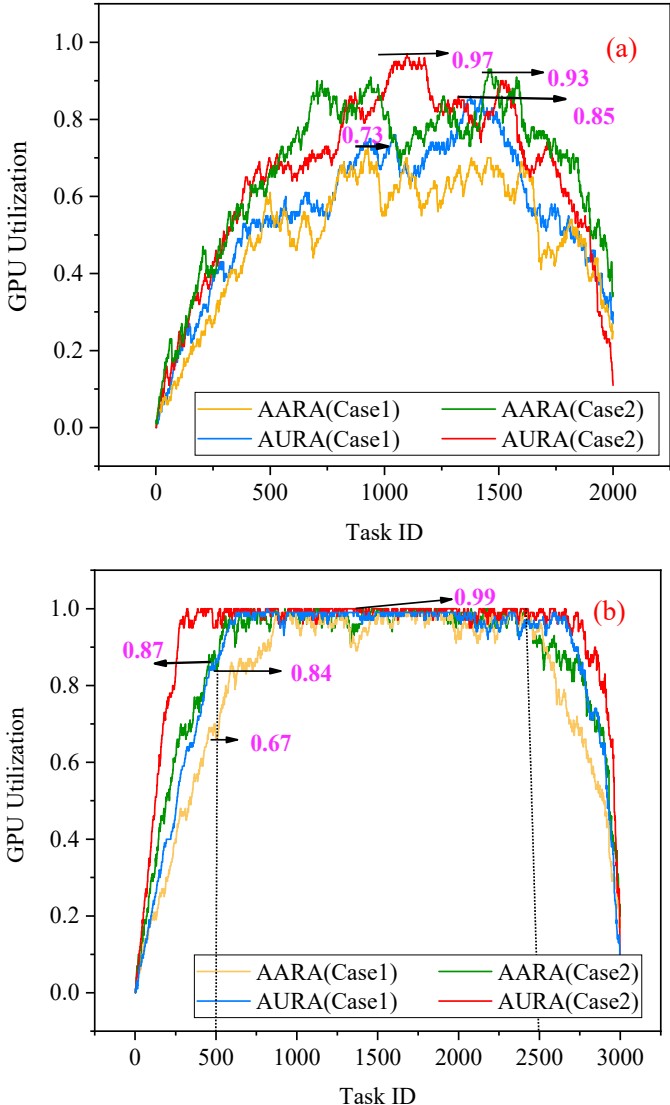

**Figure 6.** GPU utilization in different profiles of computing tasks: (**a**) 2000 tasks; (**b**) 3000 tasks.

Figure 7 shows the results of bandwidth utilization. When the task ID is between 500 and 1500, both AARA and AURA achieve peak of bandwidth utilization. The peak bandwidth utilization of AARA reaches 80% when there are 2000 tasks, and reaches 93% when there are 3000 tasks. Figure 7a shows that the peak bandwidth utilization of AARA is 5% more than AURA in Case 1, and 7% more in Case 2. In Figure 7b it can be seen that as the number of tasks increases, the bandwidth utilization increases. The maximum bandwidth utilization of AARA reaches 93%, while AURA peaks at 91%. From the overall trend, it can be observed that the AARA algorithm achieves a higher peak of bandwidth utilization compared to AURA. This is because the AARA algorithm allocates computing resources based on BCR, which may result in selecting a longer path for task deployment.

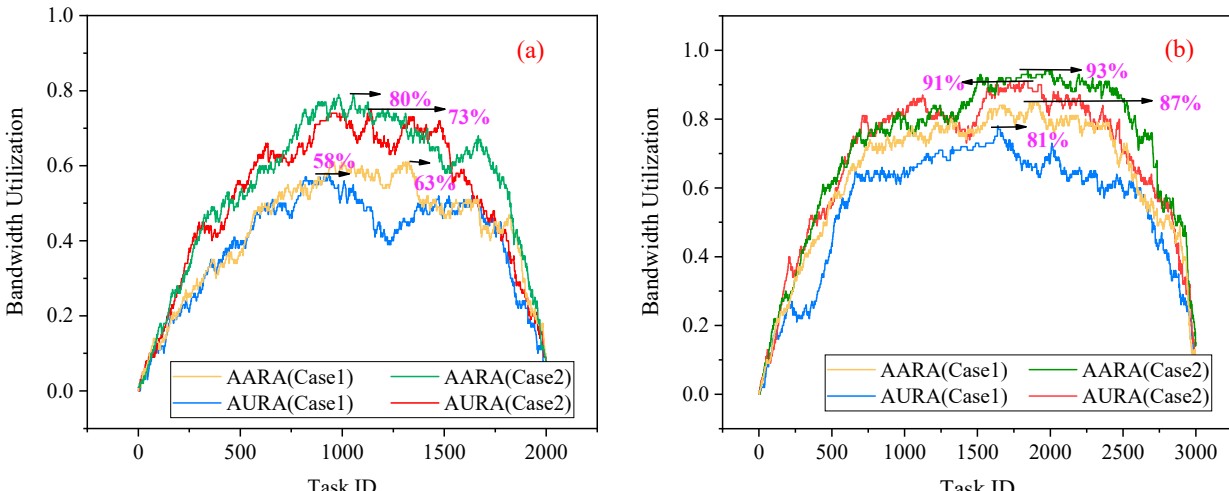

**Figure 7.** Bandwidth utilization in different profiles of computing tasks: (**a**) 2000 tasks; (**b**) 3000 tasks.

Figure 8a shows the blocking probability (BP) of two workload ratios between the CPU and GPU in 2000 and 3000 tasks. The result shows that AARA can achieve lower BP compared with the benchmark AURA with traffic loads from 100 to 600 Erlangs. The blocking ratio of the AARA algorithm is reduced by 3.7% compared to AURA in Case 2. If the traffic load and number of tasks are set as 600 Erlangs and 3000 in Case 2, respectively, nearly 8.6% of the tasks are blocked by the AURA algorithm, while for AARA, the blocking ratio is 4.9%. The reason is that the AARA algorithm considers the BCR for each task, but AURA only considers the resources that meet the requirement of the task. Thus, there is an allocation of inappropriate resource types and computing power grades for tasks, resulting in blocking of newly arrived tasks due to insufficient resources.

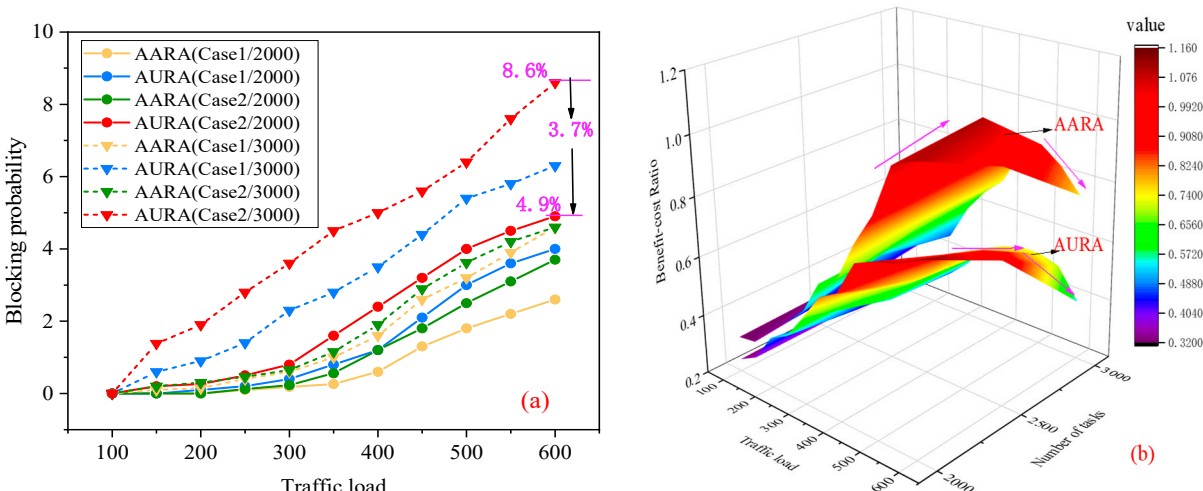

**Figure 8.** (**a**) Blocking probability in different traffic load (The number of tasks is 2000 and 3000); (**b**) BCR in different task numbers and traffic loads.

Figure 8b shows the average BCR of the tasks for different numbers of tasks and traffic loads, where the BCR of the AARA algorithm is higher than that of the AURA algorithm. In the AARA algorithm, we can see that the BCR increases with the traffic load from 100 to 450 Erlangs, but it decreases when both the number of tasks approaches 3000 and the traffic load increases from 450 to 600 Erlangs. This is because the network has limited resource capacity, and when accommodating a larger number of computing tasks, the available grades for the tasks become limited, resulting in a decrease in the benefits brought by the computing tasks. Consequently, the BCR shows a downward trend. The maximum and

minimum BCR values for the AARA algorithm are 1.1 and 0.32, respectively. The AURA algorithm has a lower BCR due to its single selection mechanism of computing power grade and lack of awareness of BCR. As the traffic load and number of tasks increase, the available computing and bandwidth resources become insufficient to accommodate more tasks. This results in a flat trend of BCR between 400 and 550 Erlangs, followed by a decrease of between 550 and 600 Erlangs.

The simulation results show that under the limited resource capacity, our proposed AARA algorithm can allocate computing resources for computing tasks by selecting the appropriate computing power grade. Meanwhile, AARA can obtain the maximum BCR and ensure the efficient utilization of heterogeneous computing resources.

## 5. Discussion

In this section, we analyze the time complexity of the AARA algorithm and AURA algorithm in Equation (11) and Equation (12), respectively. The time complexity mainly depends on calculating the k-shortest paths, finding an available wavelength channel, and searching for an optimal grade. The time complexity of the path calculation is $O(K \times |V| \times (|E| + |V| \log(|V| - 1)))$. The available free wavelength channels on the k-shortest path are calculated with the time complexity of $O(K \times N_W \times \log(|V| - 1))$. The time complexity of the AARA algorithm to search for the optimal grade is $O(K \times l)$, while for the AURA algorithm is $O(K)$. Thus, the time complexity of AARA is calculated using Equation (11):

$$
\begin{aligned}
T &= O(K \times |V| \times (|E| + |V| log(|V| - 1)) \cdot O(K \times N_W \times log(|V| - 1)) \cdot O(K \times l) \\
&= O(K^3 \times |V| \times (|E| + |V| log(|V| - 1)) \times N_W \times log(|V| - 1) \times l)
\end{aligned} \tag{11}
$$

The time complexity of the AURA algorithm is calculated using Equation (12):

$$
\begin{aligned}
T &= O(K \times |V| \times (|E| + |V| log(|V| - 1)) \cdot O(K \times N_W \times log(|V| - 1)) \cdot O(K) \\
&= O(K^3 \times |V| \times (|E| + |V| log(|V| - 1)) \times N_W \times log(|V| - 1)
\end{aligned} \tag{12}
$$

Furthermore, the limitations of and improvements of our work mainly include the following points. Firstly, our future research can expand to larger-scale simulations, including a greater variety of datasets and more evaluation metrics. Moreover, due to resource and time constraints, we may not be able to explore all possible parameter combinations in our simulations. Therefore, we have to select values within a limited range and rely on existing knowledge and experience. The complexity and diversity of the research field can also present challenges in parameter selection. Each specific problem has its unique characteristics and variables, which may require specific parameter settings. As a result, our chosen parameters may not be applicable to all situations. Secondly, we only consider the heterogeneous resources limited to CPUs and GPUs. However, there are also other computing resources such as NPUs and TPUs for different computing tasks in different scenarios. Thirdly, this work does not consider traffic prediction, which can further optimize resource allocation strategies. Finally, faults and attacks can also be considered in CPNs, as well as their impact on resource allocation and application performance. Therefore, we can further explore solutions to deal with failures and attacks in CPNs, such as establishing redundant systems and designing recovery methods for failures and attacks.

## 6. Conclusions

In this paper, we studied the resource allocation problem in CPNs with heterogeneous resources, considering three types of applications: general applications, CPU-intensive applications, and GPU-intensive applications. To quantify the efficiency of CPU and GPU utilization in CPNs, we proposed a new concept of BCR. Meanwhile, the AARA algorithm was designed to process multiple applications with limited resource capacity. With massive simulations, we analyze the different effects of AARA and AURA. In terms of blocking probability, resource utilization, and BCR, AARA achieved better performance than AURA.

The simulation results validate that more computing tasks can be accommodated by reducing 3.7% blocking probability through BCR-based resource allocation.

**Author Contributions:** Conceptualization, Y.L.; methodology, Y.L. and Y.W.; software, Y.W., J.G. and Y.F.; validation, J.G. and Y.F.; formal analysis, B.Z. and L.C.; investigation, Y.W. and B.Z.; resources, J.Z. and W.W.; data curation, W.W. and Y.W.; writing—original draft preparation, Y.W.; writing—review and editing, Y.L.; visualization, L.C. and Y.Z.; supervision, Y.Z. and J.Z.; funding acquisition Y.L. and J.Z. All authors have read and agreed to the published version of the manuscript.

**Funding:** This work was supported in part by the Beijing Natural Science Foundation (4232011), the Project of Jiangsu Engineering Research Center of Novel Optical Fiber Technology and Communication Network, Soochow University (SDGC2117), and the Fundamental Research Funds for the Central Universities and NSFC (61831003, 62021005, 62101063).

**Institutional Review Board Statement:** Not applicable.

**Informed Consent Statement:** Not applicable.

**Data Availability Statement:** Data are contained within the article.

**Conflicts of Interest:** The authors declare no conflict of interest.

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
