# Peer review of "Application-Aware Resource Allocation Based on Benefit–Cost Ratio in Computing Power Network with Heterogeneous Computing Resources"

_photonics, doi:10.3390/photonics10111273_

Round 1

Reviewer 1 Report

Comments and Suggestions for Authors

This study introduces BCR, which quantifies the CPU and GPU efficiency in CPN and the AARA algorithm for application allocation on CPN. I have the following questions about this study:

* I'm curious about the alignment of the scope of this paper with the journal. While this study pertains to computing systems, it has been submitted to a photonics journal. Is the scope of this paper appropriate for the journal?

* What is the precise definition of "cost-effective"? The authors claim that what sets this study apart from others is its consideration of cost-effectiveness, yet a clear definition is absent.

* What does "e-switch" mean? Definitions for abbreviated terms are missing.

* How are the CPUs and GPUs positioned relative to the e-switch? Are they integrated within a computer, or are they disaggregated across multiple clouds?

* How can applications be placed? Are they in the form of containers or virtual machines?

* How were the required parameters in the model obtained? The parameters in Table 1 seem incomplete.

* The evaluation section only presents the results of the authors' proposed design. There should be comparisons with other existing schemes.

* Numerous studies exist on CPU, GPU, and other resource utilizations for practical applications. It would be insightful to reference and discuss these in relation to AARA.

    * "Paleo: A performance model for deep neural networks." International Conference on Learning Representations. 2016.

    * "Prediction of the resource consumption of distributed deep learning systems." Proceedings of the ACM on Measurement and Analysis of Computing Systems 6.2 (2022): 1-25.

    * "CPU load prediction for cloud environment based on a dynamic ensemble model." Software: Practice and Experience 44.7 (2014): 793-804.

    * "Optimizing cloud data center energy efficiency via dynamic prediction of CPU idle intervals." 2015 IEEE 8th International Conference on Cloud Computing. IEEE, 2015.

    * "An ensemble CPU load prediction algorithm using a Bayesian information criterion and smooth filters in a cloud computing environment." Software: Practice and Experience 48.12 (2018): 2257-2277.

Comments on the Quality of English Language

Many parts have grammatical errors

Reviewer 2 Report

Comments and Suggestions for Authors

This paper proposed an application-aware resource allocation algorithm based on cost-benefit ratio to provide heterogeneous computing resources for different types of applications in a computing power network, and verified its performance advantages through simulation experiments. The following are the specific review comments:

(1) The authors did not clearly explain the concept and characteristics of CPN, as well as its differences and advantages from other computing networks. A more detailed introduction and definition of CPN is suggested in the Introduction section.

(2) They did not provide a specific calculation formula for BCR, but only used some parameters to represent benefits and costs. The authors are suggested to give the mathematical expression of BCR in Section V, and explain the meaning and source of each parameter.

(3) The complexity of the AARA algorithm is suggested to be analyzed, as well as it’s theoretically provement of the effectiveness and feasibility of the algorithm. The authors are suggested to add this aspect to Section V, or discuss the limitations and improvement directions of the algorithm in the Discussion section.

(4) The work did not consider possible faults and attacks in CPN, as well as their impact on resource allocation and application performance. The authors are suggested to add some simulation experiments of failure and attack scenarios in Section VI, or propose corresponding solutions in future work.

Comments on the Quality of English Language

NA

Reviewer 3 Report

Comments and Suggestions for Authors

In this paper, the concept of benefit-cost ratio (BCR) is proposed to quantify the efficiency of CPU and GPU usage in CPNs. An application-aware resource allocation algorithm is designed to handle multiple applications. Simulation results show that the algorithm has better results. The research has some significance, but still needs a lot of modifications and interpretations to meet the acceptance criteria. My specific comments are as follows:

1. the abstract describes more about the significance of the study as well as the innovativeness of this study;

2. keywords avoid using acronyms and give full names;

3. there are too many sections, it is recommended to talk about the introduction and Related Work combined in the introduction. Especially after the introduction of recent research progress, then present your own work and innovation, which is more conducive to the reader's understanding;

4. Some algorithms for parallel allocation of resources are mentioned: slime mold algorithm (10.1007/s11227-022-04599-w); Mobile edge computing (10.1109/TVT.2020.3040645).

5.l.142-143, F in Figure 1 should also be described;

6. it is suggested to integrate sections 3 and 4 into one part, since they are both introductions to the research topic of this study;

7. the pseudo-code of AARA has the problem of non-correspondence consistency of loops, which is checked by the authors;

8. suggest the authors to increase the size and number of simulation experiments to verify the effectiveness of the algorithm;

9. the conclusion part increases the innovative expression, not only the experimental results to do the description; in addition, the limitations of the algorithm and the prospect of future research also do some necessary description.

Comments on the Quality of English Language

Minor editing of English language required

Round 2

Reviewer 1 Report

Comments and Suggestions for Authors

I appreciate the authors' efforts in addressing comments from the previous round. However, I still have two remaining questions.

1. Regarding comment 6 from the previous review, the authors mentioned that the parameters were derived from either 1) previous study or 2) were determined in this study. Please provide a clear reference to the previous study for the former. For the latter, were the values selected based on empirical experiments or were they arbitrarily decided? A discussion on the limitations would further strengthen the paper.

2. Concerning comment 7, the authors stated that comparing their technique with existing ones is impossible because the latter only considers a single computing resource. Nonetheless, by contrasting with studies that focus on just one resource, this paper could highlight the limitations of previous studies that the AARA algorithm aims to improve.

Comments on the Quality of English Language

NA

Reviewer 2 Report

Comments and Suggestions for Authors

This paper proposed an application-aware resource allocation algorithm based on cost-benefit ratio to provide heterogeneous computing resources for different types of applications in a computing power network, and verified its performance advantages through simulation experiments.

The following are the specific review comments:

(1) The article did not clearly explain the concept and characteristics of CPN, as well as its differences and advantages from other computing networks. A more detailed introduction and definition of CPN is suggested in the Introduction section.

(2) The article did not provide a specific calculation formula for BCR, but only used some parameters to represent benefits and costs. The authors are suggested to give the mathematical expression of BCR in Section V, and explain the meaning and source of each parameter.

(3) The article did not analyze the complexity of the AARA algorithm, nor did it theoretically prove the effectiveness and feasibility of the algorithm. The authors are suggested to add this aspect to Section V, or discuss the limitations and improvement directions of the algorithm in the Discussion section.

(4) The article did not consider possible faults and attacks in CPN, as well as their impact on resource allocation and application performance. The authors are suggested to add some simulation experiments of failure and attack scenarios in Section VI, or propose corresponding solutions in future work.

(5) The language expression of the article is not smooth and accurate enough, and some sentences are too long or lack punctuation marks. It is recommended that the entire text be carefully proofread and revised to avoid grammatical errors and spelling errors.

(6) The structure of the article is not clear and logical enough, and some parts lack necessary transitions and connections. It is recommended that the full text be reorganized and adjusted so that there are clear connections and connections between the various parts.

(7) Some of the references of the article are not complete and updated, some documents are not very relevant to the topic of the article, and some documents are unpublished or unpeer-reviewed preprints. It is recommended to screen and supplement references and try to use officially published and authoritative documents,such as:

[1]Minimal cost server configuration for meeting time-varying resource demands in cloud centers

IEEE Transactions on Parallel and Distributed Systems 29 (11), 2503-2513               20           2018

[2]   A Game-Based Price Bidding Algorithm for Multi-attribute Cloud Resource Provision

IEEE Transactions on Services Computing   8               2018

[3]Optimizing CPU Performance for Recommendation Systems At-Scale. In Proceedings of the 50th Annual 429

International Symposium on Computer Architecture (ISCA '23)., New York, NY, USA, 17 - 21 June 2023 

Reviewer 3 Report

Comments and Suggestions for Authors

I consider that the authors have made sufficient revisions to the comments presented by the reviewers in the previous round. Therefore, I accept this paper for publication.

Comments on the Quality of English Language

English expression is OK.

Author Response

Dear Reviewer:

  Thank you for your prompt response and for accepting our paper for publication. We sincerely appreciate your time and the valuable feedback provided during the review process. Your support is greatly appreciated, and we are pleased that our revisions were found to be satisfactory. 

Round 3

Reviewer 2 Report

Comments and Suggestions for Authors

The author has made good modifications. Here are some comments.

The computational complexity of the proposed method should be discussed and evaluated.

Some references are not in the right format, it needs to be checked.

The typos should be carefully checked.
